# Personalized Benchmarking with the Ludwig Benchmarking Toolkit

**Avanika Narayan,  Piero Molino,  Karan Goel,  Willie Neiswanger,  Christopher Ré**
Department of Computer Science
Stanford University
`{avanika, pmolino, kgoel, neiswanger, chrismre}@cs.stanford.edu,`

## Abstract

The rapid proliferation of machine learning models across domains and deployment settings has given rise to various communities (e.g. industry practitioners) which seek to benchmark models across tasks and objectives of personal value. Unfortunately, these users cannot use standard benchmark results to perform such value-driven comparisons as traditional benchmarks evaluate models on a single objective (e.g. average accuracy) and fail to facilitate a standardized training framework that controls for confounding variables (e.g. computational budget), making fair comparisons difficult. To address these challenges, we introduce the open-source Ludwig Benchmarking Toolkit (LBT), a personalized benchmarking toolkit for running end-to-end benchmark studies (from hyperparameter optimization to evaluation) across an easily extensible set of tasks, deep learning models, datasets and evaluation metrics. LBT provides a configurable interface for controlling training and customizing evaluation, a standardized training framework for eliminating confounding variables, and support for multi-objective evaluation. We demonstrate how LBT can be used to create personalized benchmark studies with a large-scale comparative analysis for text classification across 7 models and 9 datasets. We explore the trade-offs between inference latency and performance, relationships between dataset attributes and performance, and the effects of pretraining on convergence and robustness, showing how LBT can be used to satisfy various benchmarking objectives.

**Code Repository**: `https://github.com/HazyResearch/ludwig-benchmarking-toolkit`

## 1  Introduction

Benchmarking has emerged as an important practice to measure progress in machine learning. Typically, benchmarking is done through leaderboards, where a participant's objective is to maximize a performance criterion on a challenging task or dataset. Prominent examples of these benchmarks include GLUE [1], SuperGLUE [2], ImageNet [3] and SQuAD [4].

As deep learning models become increasingly proficient at maximizing performance criteria like average accuracy, attention has shifted towards the need for more personalized, thorough, and thoughtful benchmarking that emphasizes a community or individual's needs [5]. In this work, we focus in particular on what we term *value-driven communities*—communities whose utility is aligned with optimizing and understanding model evaluation objectives beyond average performance. Examples of such communities include researchers interested in understanding the effects of model pretraining on robustness and industry practitioners interested in the tradeoffs between inference latency and performance. However, standard benchmarking practices carry several limitations that makes personalizing benchmarks difficult for these communities.

First, the shift to personalized benchmarks changes the nature of benchmark design, turning users into benchmark designers. The major burden on benchmark designers so far has been in formulating a challenging task, collecting and preparing data, and selecting an appropriate performance criterion to capture progress on the task. Personalization transforms this burden from managing dataset collection

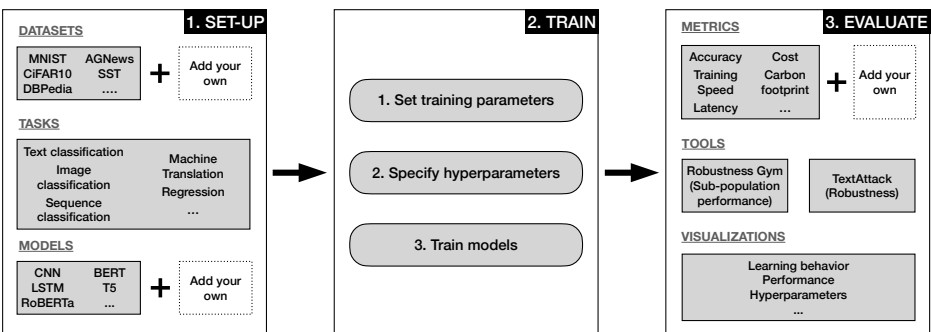

Figure 1: **Ludwig Benchmarking Toolkit.** LBT supports multi-objective evaluation, provides a standardized training framework, and includes an extensible set of datasets, models and metrics.

and curation to also managing careful training and evaluation. This shift requires new tools that permit finer-grained control over benchmarking studies, where users can customize training and evaluation based on their needs while automating as much of this burden as possible.

Second, a general goal for benchmarking is to help researchers and practitioners make apples-to-apples model comparisons and draw accurate conclusions about why some models perform better or worse. The mechanism adopted by leaderboard benchmarks limits the ability to precisely answer such questions. This is because submitted models vary substantially in their training data, compute resources, preprocessing, training protocol, and implementations [6]. These confounding factors make it difficult to draw any conclusion about what parts of an implementation were ultimately responsible for its performance, especially since these factors may matter even more than architecture differences [7, 8, 9, 10, 11].

Lastly, existing benchmarks provide relatively little utility to an individual who wants to compare a collection of models on multiple objectives such as robustness, training speed, inference latency, size, or other properties—all aspects that are of interest to the value-driven communities of researchers and practitioners that we focus on [5, 12]. Instead, leaderboards excel at catalyzing progress in the larger research community (e.g. moving from 70% to 90% accuracy on GLUE in 2 years) [13].

Taken together, these challenges highlight the need for personalized benchmarking tools that complement existing leaderboard-style benchmarks and allow researchers and practitioners in value-driven communities to (i) configure benchmark training and evaluation, (ii) fairly compare models by controlling for confounding variables, and (iii) perform multi-objective evaluation.

We take a first step in this direction and introduce the **Ludwig Benchmarking Toolkit (LBT)**, a personalized benchmarking toolkit for creating and running *configurable, standardized, and multi-objective* benchmarking studies with ease. To create a benchmark suite in LBT, users specify a task, a set of models to compare and datasets for evaluation, configure training and hyperparameter search spaces for model training, and evaluate and compare the trained models, as depicted in Figure 1. In particular, LBT has the following properties:

**Configurable.** To support configurability, LBT provides out-of-the-box support for training cutting-edge deep learning models that span classification, regression, and generation tasks across multiple modalities. LBT enables users to control training conditions by providing a simple configuration file interface for specifying training parameters and the hyperparameter search space. To support personalization, LBT makes it simple to extend the toolkit, and gives users explicit mechanisms for introducing custom models, datasets, and metrics. This is particularly useful for benchmarking models in application-specific scenarios.

**Standardized.** To enable fair comparisons between models trained using the toolkit, LBT implements a standardized training framework that ensures every model can be trained using the same dataset splits, preprocessing, training loop, and hyperparameters. During configuration, users choose which variables to hold constant across models (e.g., training time, the optimizer, preprocessing techniques, and the hyperparameter tuning budget), controlling for any potential confounders.

**Multi-objective.** To provide greater support for varied evaluation metrics, LBT expands the set of evaluation criteria beyond standard performance-based evaluation (e.g., average accuracy, F1 score) to include fiscal cost, size, training speed, inference latency, and carbon footprint [14]. Furthermore,

to enable developers to compare models on the basis of robustness and critical subpopulation performance (evaluation factors relevant for application deployment), LBT includes integrations with two popular open-source evaluation toolkits, TextAttack [15] and Robustness Gym [16].

We validate that value-driven communities can use LBT to conduct personalized benchmarking studies by performing a large-scale, multi-objective comparative analysis of 7 deep learning models across a diverse set of 9 text classification datasets. We explore hypotheses of interest to researchers and practitioners on the tradeoffs between inference latency and performance, relationships between dataset attributes and performance, and the effects of pretraining on convergence and robustness, all while controlling for important confounding factors. Our results show that DistilBERT has the best inference efficiency and performance trade-off, BERT is the least robust to adversarial attacks, and that pretrained models do not always converge faster than models trained from scratch.

## 2  Related Work

There have been several impactful works contributing to the landscape of model benchmarking. We provide a brief overview of these efforts and discuss how they relate to our work.

**Critiques of leadboard-style benchmarks.** Recently, leaderboard-style benchmarks have been critiqued extensively. Ethayarajh and Jurafsky [5] argue that existing leaderboards are poor proxies for the natural language processing (NLP) community and advocate that they report additional metrics of practical concern (e.g. model size) to enable users to build personal leaderboards. Furthermore, Rogers [6] critiques the lack of standardization in entries submitted to leaderboards suggesting that inequity in compute and data used at training time makes fairly comparing models on the basis of these reported results difficult. The aforementioned critiques are key motivations for our work.

**Flexible leaderboards.** Earlier this year, Liu et al. [17] introduced ExplainaBoard, an interactive leaderboard and evaluation software for interpreting 300 NLP models. Like LBT, Explainaboard provides tooling for fine-grained analysis and seeks to make the evaluation process more interpretable. However, it does not provide a standardized training and implementation framework that addresses the challenge of confounds when making model comparisons. Another flexible leaderboard is DynaBench [18], a platform for dynamic data collection and benchmarking for NLP tasks that addresses the problem of static datasets in benchmarks. DynaBench dynamically crowdsources adversarial datasets to evaluate model robustness. While LBT focuses on the model implementation and evaluation challenges of benchmarking, Dynabench's focus is on data curation. Most recently, Facebook introduced Dynaboard [19], an interface for evaluating models across a holistic set of evaluation criteria including accuracy, compute, memory, robustness, and fairness. Similar to LBT, Dynaboard enables multi-objective evaluation. However, Dynaboard focuses less on helping users configure personalized benchmark studies, as users cannot introduce their own evaluation criteria or datasets.

**Benchmarking deep learning systems.** Performance oriented benchmarks like DAWNBench [20] and MLPerf [21] evaluate end-to-end deep learning systems, reporting many efficiency metrics such as training cost and time, and inference latency and cost. They demonstrate that fair model comparisons are achievable with standardized training protocols, and our work is motivated by these insights.

**Benchmarking tools.** To our knowledge, there is a limited set of toolkits for configuring and running personalized benchmarking studies. ShinyLearner [22] is one such solution that provides an interface for benchmarking classification algorithms. However, ShinyLearner only supports classification tasks, a small number of deep learning architectures (e.g. does not support any pretrained language models) and only reports performance-based metrics.

## 3  The Ludwig Benchmarking Toolkit (LBT)

In Section 3.1 we describe the communities that LBT is intended to serve. In Section 3.2 we provide an overview of LBT and an example of how it is used. Lastly, in Section 3.3, we provide a more detailed discussion of the properties and features of LBT, including how LBT addresses the needs of the communities described in Section 3.1.

### 3.1 Benchmarking for Value-Driven Communities

We start by describing the users that LBT primarily targets. In particular, LBT best supports the needs of communities that satisfy the following characteristics:

1. **Value driven.** The community is aligned around objectives (e.g. training speed) for which average accuracy alone is not a good proxy. Users' goals are primarily to compare models using evaluations that align with their objectives.

2. **Prefer automation.** Users value the ability to control and configure their benchmarks, but do not want or do not know how to implement a full experimental framework from scratch.

3. **Require standardization.** Users place strong emphasis on conducting clear, standardized analyses where the training and hyperparameter optimization processes are carefully controlled, in order to advance understanding and draw accurate conclusions.

Taken together, these characteristics allow us to more precisely target communities that remain underserved by traditional benchmarks [5]. To ground this discussion, we provide examples of three communities that satisfy the characteristics we outlined:

- **ML researchers interested in performing comparative meta-analyses.** These users are researchers with extensive experience in training and evaluating ML models. Their goal is to compare models across various objectives (e.g., learning dynamics, bias, fairness, robustness, efficiency) and tease apart the effects of preprocessing, hyperparameters, and modeling choices (e.g., pretraining, model architectures) on performance. They would benefit from a standardized pipeline for training and evaluation (for fair comparison and accurate analyses), access to robust training metadata and evaluation metrics, and tooling to perform fine-grained evaluation. Since these users are experts, they require explicit mechanisms for customization (e.g., custom datasets, models and metrics).

- **Industry practitioners interested in deployment-readiness.** These users are engineers with low to medium experience training and evaluating ML models. Their goal is to find the best model for their task of interest as quickly as possible, taking into account deployment-specific criteria such as inference latency and training speed. They would benefit from a simple user-interface that removes the need to write any deep learning code, provides extensive reporting of metrics, and provides out-of-the-box support for common ML tasks and architectures. They value the ability to add application specific datasets and evaluation criterea to their benchmark study.

- **Subject-matter experts interested in task-specific performance.** These users are domain experts (e.g. cardiologists) with limited experience training and evaluating deep learning models. Their goal is to find the best model for their task of interest based on performance on domain-specific data (e.g. ECG data for arrhythmia classification) and specific error metrics. Similar to the previous example, they would prefer a low-code, simple user interface and necessitate configurability to introduce domain-specific datasets and metrics.

In the following sections, we describe LBT's *configurable, standardized, and multi-objective* toolkit, and show how it can enable value-driven communities to create personalized benchmark studies.

### 3.2 Toolkit Overview and Usage

First, we provide a brief overview of LBT and describe how it is used. LBT enables users to run end-to-end model benchmarking studies and is composed of four main components: off-the-shelf task, model, and dataset support, model training, evaluation, and a shared research database. Users can choose from a large number of tasks, train models with a pipeline that provides standardization (e.g., of preprocessing, training loops, and hyperparameter searches), evaluate results across objectives of interest, and publish benchmark outcomes to a shared Elasticsearch database. All components in this toolkit are configurable from a set of simple files.

Running benchmarking experiments in LBT is an easy five-step process. In each experiment, users populate three configuration files: one for their task, model, and hyperparameter space. We describe these configuration files and the five-step process next.

**1. Define** the experiment: Users can choose from any of the supported tasks, models, and datasets already available in LBT or easily add their own. Each supported task has an associated task config file that specifies the end-to-end model structure corresponding to the task.

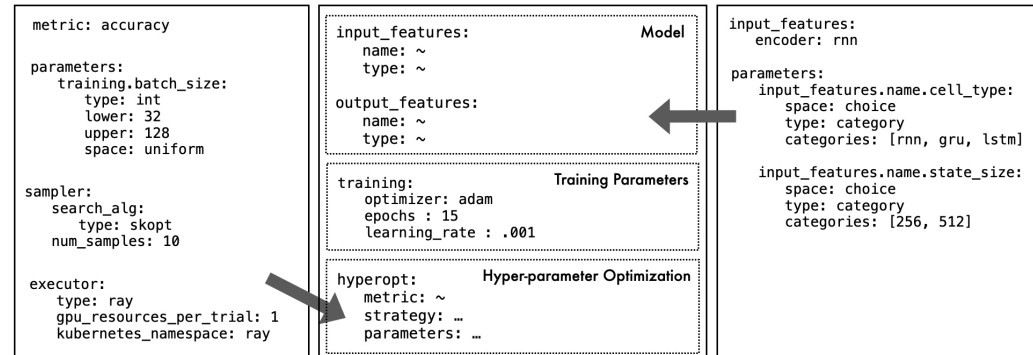

Figure 2: **Sample LBT configuration files.** Setting-up an experiment in LBT requires populating 3 configuration files that define task, model / training parameters and hyperparameter optimization.

*Example: Consider a text classification experiment comparing the performance of an RNN and the ELECTRA model. We will run the experiment across 2 datasets: Social Bias Frames and Hate Speech and Offensive Language. Figure 2 (center) presents a sample task config file.*

**2. Specify** the training parameters and hyperparameter search: Users specify values for the hyperparameters that should be constant across all training runs, as well as the hyperparameters they would like to optimize.

*Example: We specify model-specific parameters and their search space in the model config files (Figure 2; right). To control for the optimizer, learning rate, and training epochs, we set their values to "adam", "0.0001", and "15" in the task configuration file. We specify our optimization metric (validation accuracy), parameters we would like to optimize (batch size) and our search algorithm (skopt) in the hyperparameter config file (Figure 2; left).*

**3. Run** the optimization experiment: Running an experiment is a simple one-line command. When an experiment is run, a configuration file for each task, model, and dataset combination is saved (See Figure A.4 for an example). The saved file records the model architecture, training variables, and hyperparameter settings and can be used to reproduce an experiment with a single command: `python experiment_driver.py -reproduce <path to experiment config file>`

*Example: We choose to run our experiment on a GCP cluster across four machines. We specify our compute environment by passing in a flag at runtime and by specifying the name of our Kubernetes cluster in the hyperparameter configuration file (Figure 2; left).*

**4. Evaluate** the results: Users can perform an in-depth meta-analysis using the set of performance-based evaluation metrics recorded by Ludwig during model training, along with the additional metrics logged by LBT. On top of analyses performed using the reported metrics, users can use the TextAttack and Robustness Gym APIs to better understand fine-grained aspects of model performance.

*Example: We want to gain a better understanding of bias in our offensive language detection classifiers. We test if these models classify text with African American Vernacular English (AAVE) as offensive more frequently than without [23]. We define a Robustness Gym slice for samples containing words unique to AAVE, and compare model performance on this subpopulation to identify bias.*

**5. Publish** the experiment: All experiments run using LBT can be uploaded to a shared Elasticsearch research database. Due to the flexible nature of Elasticsearch, users can update experiments with any additional metrics and analyses over the duration of their study.

*Example: To publish results to the database we add one command-line flag at runtime:* `python experiment_driver.py ... -esc elasticsearch_config.yaml`

### 3.3 Toolkit Design and Features

Next, we describe the key design choices and features of LBT which enable value-driven communities to create personalized benchmarks.

**Configurable**

*Out-of-the-box support for tasks, datasets, and models.* LBT integrates directly with the popular *Ludwig Deep Learning Toolbox (Ludwig)* [24], enabling LBT to use the existing models, datasets, and hyperparameter tuning methods available in Ludwig. Thus, LBT can support several tasks out-of-the-box like multi-class and multi-label classification, regression, sequence tagging, and sequence generation over a diverse set of input data types such as tabular, image, text, audio, and time series.

*Simple user-interface for configuration.* Configuring a benchmark study in LBT is as simple as specifying a task (e.g. image classification), choosing a set of models to compare from the ones available (or implementing a new one), selecting datasets for evaluation and declaring training parameters. This is achieved by populating declarative configuration files for the benchmark task, training parameters, model-specific parameters, and hyperparameter search space (see Figure 2).

*Extensible to new tasks, datasets, models and evaluations.* LBT provides explicit mechanisms for users to personalize and extend the toolkit to their needs. Figure A.1 illustrates how to register a new dataset and custom evaluation metric. Adding new models and tasks is simple, and requires implementing a new instance of an encoder or decoder in Ludwig (functions mapping from input data to hidden representation and from hidden representation to predictions respectively).

**Standardized**

*Standardized model training.* To ensure that models trained in LBT can be compared fairly, LBT includes a standardized framework for training and hyperparameter optimization. Using this framework, models can be trained using the same dataset splits, preprocessing techniques, training loop, and hyperparameter search space if necessary. LBT harnesses the extensive hyperparameter tuning support in Ludwig to provide automated hyperparameter optimization when training benchmark models. LBT supports running distributed, multi-node experiments both locally or on remote clusters such as Google Cloud Provider (GCP), Amazon Web Services (AWS), and SLURM.

*Shared research database.* To support communities in sharing, replicating, and extending experiments, we provide access to a shared research database that stores the results, reported metrics, and metadata of experiments run in LBT. Experiments are uploaded to the database along with their configuration files. Users can search the database to view and download experiments run by other users, and reproduce them using the experiment's configuration file.

**Multi-Objective** LBT exposes three flavors of evaluation support: metrics, tools, and visualizations.

*Metrics for multi-objective evaluation.* With respect to metrics, LBT expands the scope of traditionally reported evaluation metrics (e.g. average accuracy) to include cost, efficiency, training speed, inference latency, model size, and more. Table A.1 details the additional metrics supported in LBT.

*Integrations for fine-grained evaluation.* To further support custom evaluations, LBT enables users to compare models based on robustness. In this work, we define robustness as critical subpopulation performance [16] and sensitivity to adversaries and input perturbations [15, 25], acknowledging that this is not a universal definition of robustness as other dimensions of robustness exist (e.g. robustness to online distributional shift). Nonetheless, LBT integrates with two open-source evaluation tools for measuring robustness: Robustness Gym (RG) [16] and TextAttack [15]. LBT's API for RG lets users inspect model performance on a set of pre-built subpopulations (e.g., sentence length, image color etc.), as well as add more subpopulations for their data and use cases (see Figure A.2 for an example). The TextAttack integration helps LBT users evaluate model robustness to input perturbations (see Figure A.3 for sample API usage.

*Visualizations.* Finally, LBT provides an API to generate visualizations for learning behavior, model performance, and hyperparameter optimization, using statistics generated during model training.

## 4    Case Study: Large-Scale Text Classification Analysis

Next, we demonstrate how users with diverse benchmarking objectives can configure personalized benchmarks and conduct deep, comparative meta-analyses using LBT. The goal of this case study is twofold. First, we seek to show that when using LBT we can replicate previously reported experimental results accurately. Second, we want to demonstrate how LBT can address the unmet needs of value-driven communities in running configurable, standardized, and multi-objective benchmark studies. As such, the goal is not to show novel insights into models but rather to demonstrate the

Table 1: **Overall Performance**. The table reports the accuracy of the top performing models for each dataset and model pair.

| Model | Dataset | | | | | | | | |
|---|---|---|---|---|---|---|---|---|---|
| | HS | AG | SST5 | MGB | IR | GE | YR | DBP | SBF |
| RNN | 0.875 | 0.910 | 0.476 | 0.879 | 0.769 | 0.458 | 0.954 | 0.986 | 0.653 |
| Stacked Parallel CNN | 0.883 | 0.911 | 0.468 | 0.883 | 0.753 | 0.448 | 0.948 | 0.986 | 0.640 |
| DistilBERT-base | 0.915 | 0.934 | 0.528 | 0.888 | 0.758 | 0.549 | 0.965 | 0.991 | 0.675 |
| BERT-base | 0.919 | 0.943 | 0.530 | 0.892 | 0.801 | 0.546 | 0.969 | 0.992 | 0.687 |
| ELECTRA-base | 0.911 | 0.932 | 0.540 | 0.896 | 0.747 | 0.542 | 0.969 | 0.990 | 0.663 |
| T5-small | 0.912 | 0.935 | 0.541 | 0.894 | 0.769 | 0.535 | 0.968 | 0.991 | 0.680 |
| RoBERTa-base | 0.918 | 0.940 | 0.551 | 0.898 | 0.780 | 0.541 | 0.973 | 0.991 | 0.687 |

practicality and usability of the toolkit. With these goals in mind, we use LBT to conduct a large-scale text classification benchmark study that spans 4 tasks, 9 datasets, and 7 models with a total of 1260 trained models, all evaluated across a variety of metrics. To ground our benchmarking, we use the metrics and tools supported by LBT to study a few relationships in particular: the tradeoff between efficiency and performance, effects of dataset attributes on performance, and the effects of pretraining on performance. We provide experimental details in Section 4.1 and describe our hypotheses and findings in Section 4.2.

## 4.1 Experimental Setup

We conduct benchmarking on text classification tasks due to the abundance of available datasets, models suitable for the task and published results [26, 27, 28].

**Datasets.** We chose the following nine classification datasets: Hate Speech and Offensive Language (HS) [29], AGNews (AG) [30], DBPedia (DBP) [30], Yelp Review Polarity (YR) [30], SST5 (SST5) [31], MD Gender Bias (MGB) [32], Irony Classification (IR) [33], GoEmotions (GE) [34] and Social Bias Frames (SBF) [35]. The datasets were chosen based on their diversity in average sentence length (ranging from 5 to 132), dataset size (ranging from 1364 to 56000), number of classes (ranging from 2 to 27), and language type (e.g. formal vs. informal language). The datasets cover four common text classification tasks: sentiment analysis, emotion classification, topic classification, and hate and offensive speech detection, and span both binary and multi-way classification.

**Models.** We analyze five pretrained language models and two text encoders trained from scratch. The pretrained models are BERT-base [36], DistilBERT-base [37], Electra-base [38], RoBERTa-base [39], T5-small [40] and were chosen due to variance in size and pre-training strategies. The text encoders are a stack of bidirectional RNN layers (with the cell type chosen to be RNN, Long Short-Term Memory layer (LSTM) [41], or Gated Recurrent Unit (GRU) [42]) and a stacked implementation of the CNN for sentence classification [43] (Stacked Parallel CNN or SP-CNN).

**Hyperparameters.** Across all experiments we use the Adam optimizer [44] and the Scikit Optimize (skopt) hyperparameter search algorithm [45], sampling 20 hyperparameter settings per dataset and model pair (e.g. BERT and SST5). We optimize over learning rate, model hidden dimension size, and model-specific parameters such as cell type for the RNN model and size and number of stacked layers for the SP-CNN. All experiments used Tesla T4 GPUs on GCP.

## 4.2 Results and Analysis

**Average Accuracy Analysis.** First, we compare models on the basis of average accuracy to demonstrate that standard, accuracy-based benchmark comparisons are possible in LBT. Table 1 shows the performance of the best hyperparameter configuration for each model-dataset pair. We verify that these results are aligned with previously reported experimental results [27, 28]. Based on average accuracy alone, RoBERTa-base and BERT-base have the best performance across all nine datasets. We note that on some datasets, the accuracy gap between the best and worst model is as large as 0.09 (GE, SST5) while only 0.02 on others (MGB, DBP).

**Value-driven Analysis.** Next, we aim to validate the efficacy of LBT in enabling value-driven communities to create personalized benchmark studies. We do this by structuring our analysis into three themes: efficiency and performance tradeoffs, effects of dataset attributes on performance, and

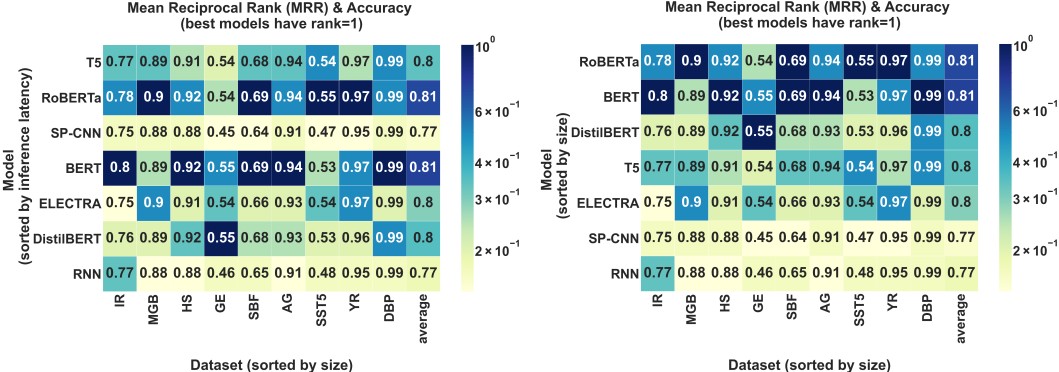

(a) Model size and inference latency are not directly correlated.

(b) Larger models perform better than smaller ones, regardless of dataset size.

Figure 3: **Mean Reciprocal Rank & Accuracy.** In (a) and (b) numbers are accuracy scores and the colors represent the MRR of a model for each dataset (darker indicates better performance).

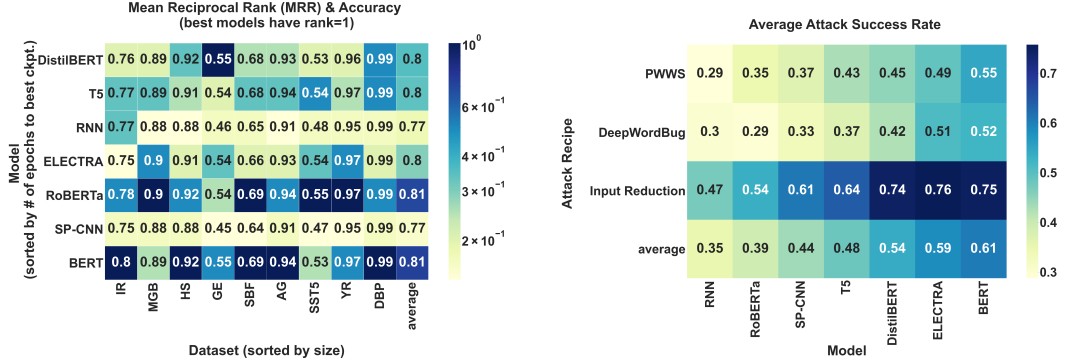

(a) DistilBERT and T5 take the longest to converge.

(b) BERT and ELECTRA are the least robust to adversarial attacks.

Figure 4: **Effects of Pretraining**. In (a), the numbers are accuracy scores and the colors represent the MRR of a model for each dataset (darker indicates better performance). In (b), the numbers are the average attack success rate of an attack strategy across all datasets.

effects of pretraining on performance. We use LBT to test multiple hypotheses related to these themes. The proposed hypotheses and associated analysis demonstrate how users with various objectives can effectively use LBT to achieve their benchmarking objectives.

*1. Inference Latency and Performance Tradeoffs*: For an engineer looking to deploy a text-classification model in production, comparing models based on their size and inference efficiency is of significant interest as low latency is critical to delivering real-time, inference-based services. To demonstrate that LBT supports such a benchmark comparison, we investigated whether there is a trade-off between performance and latency and if better-performing models, which are typically larger, have slower inference speeds. In Figure 3a, we see that BERT, which obtains the best performance on the largest number of datasets, has lower latency than RoBERTa and T5-small, suggesting that inference efficiency and performance are not directly related. Our results also indicate that DistilBERT has a very convincing tradeoff between inference speed and performance.

*2. Dataset Attributes and Performance*: For practitioners trying to find the best model for their datasets, it is useful to better understand how model performance differs as a function of dataset attributes such as number of samples or average sentence length. We show how LBT can be used to understand these relationships by testing the following hypotheses. Based on existing works in the literature [46], we hypothesized that simpler models would perform better on smaller datasets as they overfit less. Figure 3b indicates that larger models outperform the smaller, simpler models across all datasets. Furthermore, we hypothesized that evaluating on smaller datasets would result in the most variance in model performance. However, Figure 5b shows that the datasets with the highest

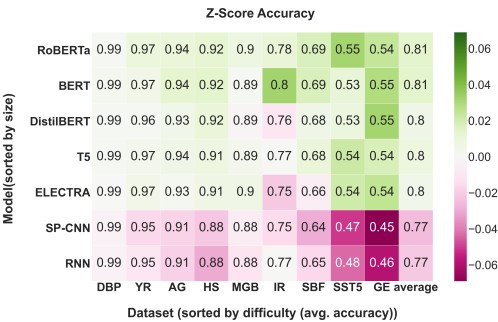

(a) Datasets which are hardest to learn have the greatest variance in performance.

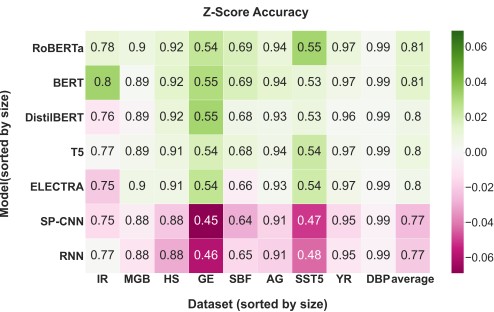

(b) Variance in performance is greatest for midsize datasets, not small datasets.

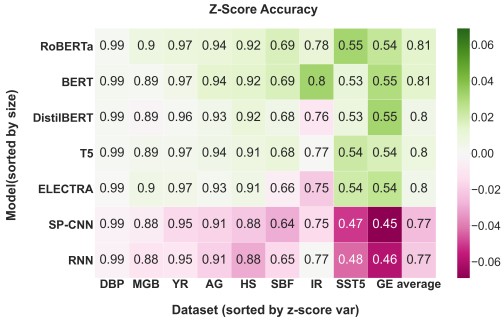

(c) Variance in performance across models is greatest for GE and SST5.

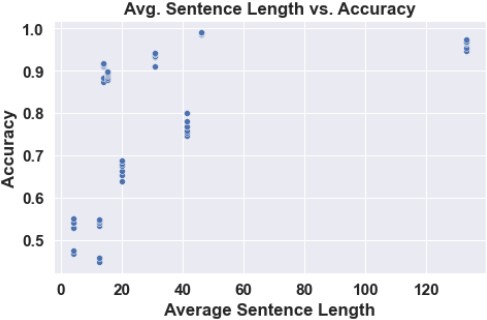

(d) Model performance is positively correlated with sentence length.

Figure 5: **Dataset Attributes and Performance**. In (a), (b), and (c), the numbers are accuracy scores and the colors represent z-score.

variance are the midsize ones. This is contrary to common belief that pretrained models have a greater advantage over models trained from scratch in data-constrained settings [47]. Figure 5a illustrates that the datasets with the highest variance in performance (where pretraining provides the biggest advantages) are those that are the most difficult (based on average accuracy). Lastly, we hypothesized that performance is positively correlated with sentence length. Figure 5d suggests that there is a positive correlation between average sentence length and performance, confirming our hypothesis.

*3. Effects of Pretraining*: For a researcher trying to better understand the effects of pretraining, exploring the impact of pretraining on robustness and model convergence is an interesting research direction [48, 49]. We demonstrate how the features of LBT make exploring the aforementioned relationships feasible. Inspired by prior work on the impact of pretraining on robustness and model convergence [48], we hypothesized that (i) pretrained models are more robust to adversarial attacks and that (ii) pretrained models should converge faster than models trained from scratch [49].

To test (i), we used the LBT TextAttack integration to compare the robustness of models to three different types of attack. The three attacks we used were DeepWordBug [50] (character insertion, swap, deletion, and substitution), PWWS [51] (synonym swap), and Input Reduction [52] (word deletion). As shown in Figure 4b, we see that the average successful attack rate is high for fine-tuned BERT models which suggests that these models are less robust to input perturbations. In contrast, RoBERTa and the RNN model have surprisingly low attack success rates. These results disprove our hypothesis and warrant further analysis.

To test (ii), we used the number of epochs elapsed until the best checkpoint as a proxy metric for model convergence. Figure 4a shows that some pretrained models do indeed converge fast (BERT and RoBERTa), while others (T5 and DistilBERT) are actually the slowest to converge.

## 5    Limitations and Conclusion

**Limitations.** We begin by acknowledging the limitations of LBT. First, the standardized training framework used to run experiments in LBT results in a trade-off between making fair comparisons on a limited model input space and making inaccurate comparisons on an unconstrained model

input space. Second, while dataset curation is a key challenge in constructing benchmark studies, LBT doesn't currently provide tooling for curating robust and comprehensive evaluation datasets. Finally, we identify LBT's dependence on Ludwig for task, model, and training support as a potential drawback. Currently, Ludwig focuses on supervised models and does not support tasks such as question answering or summarization. However, Ludwig is a growing platform supported by an active developer community, so expanded support for new tasks is likely. Moreover, while Ludwig is designed to be extensible to new models, doing so could be time consuming (take on the order of a few hours) and might serve as a potential bottleneck for users who want to spin up a benchmark study on a more expedient timeline.

**Conclusion.** In this work, we present LBT: an extensible toolkit for creating personalized model benchmark studies across a wide range of machine learning tasks, deep learning models, and datasets. We demonstrate how LBT helps value-driven communities more appropriately benchmark models by (i) providing configurable interface for creating custom benchmarks studies, (ii) implementing a standardized training framework that helps users study the tradeoffs and effects of the variables they care about by controlling for confounding variables, and (iii) providing access to a diverse set of evaluation metrics useful for multi-objective evaluation.

# 6 Ethical Considerations

We acknowledge that there are several ethical considerations pertaining to our work. Firstly, as a benchmarking toolkit, we make use of a variety of open-source datasets. In some cases, we have limited knowledge as to how the datasets were curated and if the data was collected in an ethical manner [53, 54]. Moreover, these datasets can contain several harmful biases (e.g., gender, race) that can be further propagated by models trained on their contents [53]. Another concern is data poisoning, where datasets are tampered with the intent of biasing a downstream trained model [55]. Secondly, LBT makes use of several pretrained language models. An abundance of recent work has highlighted a variety of biases that exist in these models [56, 57]. That being said, because the toolkit is modular, users have the agency to replace any datasets and models in their benchmark studies that they believe might have ethical issues. Moreover, LBT also provides the community with the necessary tools to compare models and datasets based on bias. We hope that the community will use LBT for these objectives.

## Acknowledgments and Disclosure of Funding

We are thankful to Michael Zhang, Laurel Orr, Sarah Hooper, Dan Fu, Arjun Desai and many other members of the Stanford AI Lab for helpful discussions and feedback. We would also like to thank Richard Liaw (Ray) and Travis Addair (Horovod) for their support and guidance in building LBT. We gratefully acknowledge the support of NIH under No. U54EB020405 (Mobilize), NSF under Nos. CCF1763315 (Beyond Sparsity), CCF1563078 (Volume to Velocity), and 1937301 (RTML); ONR under No. N000141712266 (Unifying Weak Supervision); ONR N00014-20-1-2480: Understanding and Applying Non-Euclidean Geometry in Machine Learning; N000142012275 (NEPTUNE); the Moore Foundation, NXP, Xilinx, LETI-CEA, Intel, IBM, Microsoft, NEC, Toshiba, TSMC, ARM, Hitachi, BASF, Accenture, Ericsson, Qualcomm, Analog Devices, the Okawa Foundation, American Family Insurance, Google Cloud, Salesforce, Total, the HAI-AWS Cloud Credits for Research program, the Stanford Data Science Initiative (SDSI), and members of the Stanford DAWN project: Facebook, Google, and VMWare. The Mobilize Center is a Biomedical Technology Resource Center, funded by the NIH National Institute of Biomedical Imaging and Bioengineering through Grant P41EB027060. The U.S. Government is authorized to reproduce and distribute reprints for Governmental purposes notwithstanding any copyright notation thereon. Any opinions, findings, and conclusions or recommendations expressed in this material are those of the authors and do not necessarily reflect the views, policies, or endorsements, either expressed or implied, of NIH, ONR, or the U.S. Government.

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
