# OpenReview forum: "Personalized Benchmarking with the Ludwig Benchmarking Toolkit"
_NeurIPS.cc/2021/Track/Datasets_and_Benchmarks/Round1 — NeurIPS 2021 Datasets and Benchmarks Track (Round 1)_

### Official Review · Reviewer_CjLP · 2021-06-30
**A solid contribution to this conference**

**Rating:** 8
**Confidence:** 4

**Strengths:**

This is a useful tool for researchers, and is directly aligned with the spirit of the NeurIPS benchmarks and datasets track. This benchmarking framework is specifically designed to make it easier for small research groups (who  lack the resources afforded by well-funded teams and training hardware found at large organizations) to carry out research in machine learning in a more principled and reproducible manner. Thus, it is directly aimed at making at making machine learning research more equitable and accessible.

The authors provide relevant code and configuration file examples that demonstrate how to use the benchmarking framework. The authors' choice to leverage open-source software infrastructure already in common use as a foundation for their framework is a wise one, and increases the likelihood that the ideas in this paper will stand the test of time.

Even if the authors' actual code is not directly used by other other researchers, the ideas and challenges outlined in the paper will provide useful insight to others who wish to build on them. In particular, I think the focus on easy user-configurability and the emphasis on measuring computational resource usage as well as accuracy is important. I hope other common machine learning benchmarking software takes note of these ideas.

The code is freely available, and is of decent quality as far as research code goes.

**Weaknesses:**

No major weaknesses that are not already accounted for in the paper. In fact, the authors include a refreshingly up-front discussion of the limitations of their own work.

In particular, the authors' reliance on other software for implementing the benchmarking limits the applicability of the framework primarily to supervised models and a limited selection of datasets. In addition, this paper does little to help users address the thorny tradeoff the accuracy and conclusions drawn from the benchmarks vs. the increased computational resources required to explore more of the parameter space.

Both of these points are addressed in the paper.

**Additional Feedback:**

The following would enrich the paper overall:
* A brief discussion of how other benchmark makers (gym, etc) might incorporate lessons learned over the course of this work into their libraries.
* Remove filler words like "easy", "simple", etc.

**Clarity:**

The paper writing and layout is excellent. The authors clearly outline their goals and target audience, and the potential uses of this framework.  They provide code examples in the paper along with experimental results.

**Correctness:**

The authors use their framework to run an ensemble of text classification experiments. They train seven different models on nine different datasets on a variety of hyperparameters. They have several interesting findings, including some evidence that pre-training models does not have as great an effect on model performance as is commonly believed. The experiments are an appropriate test for the benchmarking framework.

**Documentation:**

Code is linked to in the paper and the submission. The software well-documented and the code is fairly readable. The datasets are standard, as are the models used.

**Ethics:**

I see no major ethical concerns with this paper.

**Relation To Prior Work:**

This authors provide a very good discussion of related benchmarking software, and where their framework fits in context. In particular it emphasizes that this paper is more focused on enabling multi-objective benchmarks that can be reproduced easily. The present work is focused on giving holistic measures of performance, both predictive and computational, as opposed to merely predictive measures.

**Summary And Contributions:**

The authors introduce an extension to a popular open-source framework. This framework, Ludwig, allows users to easily build and train models by specifying simple configuration files. The authors' extension of this framework allows for researchers to specify a set of benchmarking tests for these their own models.

The authors' benchmarking framework allows users evaluate the effects of choosing different hyperparameters, optimizers, hardware, allotted training time and model on several relevant objectives. These objectives comprise both model performance (e.g. accuracy, robustness) and computational resource usage and characteristics (e.g. cost, training speed, latency). The value of this framework comes from allowing users to perform these tests by simply specifying the desired datasets, models, metrics and objectives in configuration files mirroring those already in general use.

---

> ### Author Response · Authors · 2021-07-12
> **Response to Reviewer #3**
>
> Thank you for your review, noting that LBT is a "useful tool for researchers", "directly aligned with the spirit of the NeurIPS benchmarks and datasets track", allows for carrying out research in a "more principled and reproducible manner", and overall makes machine learning research more "equitable and accessible". Moreover, your positive comments reinforcing the benefits of the design choices of the toolkit, namely our choice to leverage open-source software and our configuration-based interface, were particularly encouraging. Below, we address your comments and suggestions, while pointing out corresponding updates to the revised manuscript (additions are made in blue).
>
> 1. **Support for resource-constrained users**
>     We agree that helping users address the tradeoff between the quality of conclusions drawn from benchmarks and the increased resources required to explore more of the parameter space would be very valuable. We currently support resource-constrained LBT users by facilitating access to hyperparameter scheduling algorithms such as hyperband and population-based training techniques which are more resource-efficient. While we acknowledge that this is not a complete solution, we see it as a first step in the right direction. Furthermore, we hope that the experiments gathered in the shared database might inform resource-constrained users as to the optimal seed hyperparameters and relevant search spaces for their given datasets and models.
> 2. **Takeaways for other benchmark creators and toolkit maintainers**
>     Thank you for suggesting that we include a discussion as to how other benchmark creators can incorporate the lessons learned from LBT into their libraries. In response to this feedback, we have added a new Appendix section, "A.7 Lessons Learned", that contains such a discussion.

---

> > ### Comment · Reviewer_CjLP · 2021-07-20
> > **Thank you**
> >
> > Excellent! Thank you for taking the time to incorporate this feedback. I think those are great lessons to take away, and apply more generally to any research/software engineering project.

---

### Official Review · Reviewer_2wKV · 2021-07-01

**Rating:** 5
**Confidence:** 2
**Correctness:** Evaluation seems reasonable and correct
**Clarity:** Paper is clear

**Strengths:**

- I have the impression that the toolbox itself can be very useful
- The paper is mainly well written and easy to understand but often I would want to have additional/more precise information

**Weaknesses:**

The main weakness of the paper is the lack of insight from the case study.
Ideally in a benchmark paper I would prefer to gain additional knowledge. Unfortunately when I read this paper I do not get the feeling that I learned something new.

On top of that several aspects are not clear to me:
- The paper claims to support latency/cost measurements. it is not clear to me for which devices this is possible or how this is implemented. Are these hard measurements/simulators/analytical models or are hooks present for the users to provide this information.
-It would be good to provide an overview of the models that are already available in the toolbox.

**Additional Feedback:**

N/A

**Documentation:**

This seems fine

**Relation To Prior Work:**

The paper re-uses existing datasets. it is unclear which insights coming from the combined evaluation are truly novel. This is the main weakness of the paper

**Summary And Contributions:**

The paper provides a toolkit for benchmarking models in a uniform way. To enable this they include infrastructure to standardize model training and and hyper parameter optimization.
The paper also claims to include the ability to use it for multi objective evaluations.  This includes cost/ efficiency latency training speed etc ...

The paper then uses the framework to perform a case study. Here the paper re-uses existing datasets to benchmark classifiers. And in the case study they use 5 pre-trained models and 2 models trained from scratch.
- They show that the a higher latency model does not always perform better than a lower latency model (But not information on how the latency was measured is in the main paper).
- They show that there is not a single model that is best for all tasks.
- They provide results on robustness for different models but no in depth analysis
- They give results that shows it is hard to predict which models will converge further.

I am very conflicted to give this paper a score. On the one hand, there is nothing fundamentally wrong with the paper. On the other hand, there is no case study that provides new insight. This makes the paper sit right on the edge for me. I could agree with acceptance and rejections.

---

> ### Author Response · Authors · 2021-07-12
> **Response to Reviewer #2**
>
> Thank you for your thoughtful comments and suggestions. Below, we address each comment in turn, while pointing out updates to the revised manuscript (additions in blue).
>
> 1. **Lack of insight provided by the text classification case study**
>     Thank you for highlighting your concern about the lack of insights provided by the case study. We first describe the goal of the case study and why we believe the chosen experiment and accompanying analysis satisfies that goal. In doing so, we hope to better align readers’ expectations as to what insights they should anticipate to gain from the case study. We have made similar additions to Section 4 "Case Study: Large-Scale Text Classification Analysis" in the manuscript to help other readers better contextualize the case study.
>
>     Our main contribution is the novel LBT toolkit for creating personalized benchmark studies. The goal of our case study is to validate that LBT can (1) be used to replicate previously reported experimental results accurately and (2) address the unmet needs of value-driven communities (see *S3.1* for definition of value-driven communities) in running configurable, standardized, and multi-objective benchmark studies. *Our goal was not to show novel insights into models but rather demonstrate the practicality and usability of the toolkit.*
>
>     For (1), we choose a standard task—text classification—with many existing datasets and baselines with which we validate the correctness of our results (*S4.2-Average Accuracy Analysis*).
>
>     For (2), we demonstrate how value-driven communities can easily interact with LBT to run configurable, standardized, and multi-objective benchmarking.
>     - **Configurable**: we configure LBT to compare a diverse set of 9 datasets and 7 models.
>     - **Standardized**: we define variables to be held constant across models, declare training parameters and the hyperparameter search space, and finally train models using the same dataset splits, preprocessing, training loop, and hyperparameters (See *S4.1* for the hyperparameter search used, *Appendix A.3* for configuration files).
>     - **Multi-objective**: we structure the experiment analysis through the perspective of three different value-driven communities (*S4.2-Value-driven Analysis*) with varying objectives (e.g., inference speed, model size, robustness). Our analysis shows that these users can effectively and easily use LBT to carefully compare models.
>
>   We hope that this clarifies that the goal of the case study was to demonstrate the merits of LBT in enabling value-driven communities to conduct empirically valid, custom benchmark studies rather than to gain novel insights on text classification.
>
>   Lastly, while our case study is not designed to target new insights into text classification models, our case study includes new results regarding the effects of pretraining on robustness and model convergence. For instance, we find that BERT is less robust to input perturbations than models trained from scratch, which contradicts prior work that pretrained models have better generalization [1] (*S4.2-Effects of Pretraining*).
>
> 2. **More precise information regarding metrics and models**
>     Thank you for pointing out the need to provide a more thorough explanation as to how inference latency and cost are measured in LBT. In the revised manuscript, we have added additional details to the appendix section, "A.1 Metrics", that describes how metrics are computed. In an additional section, "A.3 Off-the-shelf Models", we include models that are available out-of-the-box in the toolkit. Below we summarize these clarifications.
>
>     In LBT, inference latency is computed as the average time (measured using the Python datetime module) for a model to perform inference on one data sample. The average time is approximated using 25 random samples from the test set.
>
>     For cost, we measure the total dollar spent per trained model. This value is computed using the hourly public cloud instance prices for the machines used in the experiment.
>
>     The model architectures currently supported in the toolkit include RNNs, CNNs (both stacked and parallel), LSTMs, Transformers, TabNet, ResNet, MLP-Mixer, most of the pretrained language models available in HuggingFace such as BERT, RoBERTa, ELECTRA, DistilBERT, XLNet, T5, XLM, GPT, GPT-2, ALBERT, FlauBERT, CamemBERT, CTRL, XLM-RoBERTa and Longformer. Because of Ludwig’s extensibility, adding an additional model is straightforward, as described in Appendix "A.3 Off-the-shelf Models".
>
>
> Citations:
>
> [1] Hendrycks, D., Lee, K., & Mazeika, M. (2019, May). Using pre-training can improve model robustness and uncertainty. In International Conference on Machine Learning (pp. 2712-2721). PMLR.

---

> > ### Comment · Reviewer_2wKV · 2021-07-17
> > **Final comments**
> >
> > Thanks for the clarifications.
> >
> > Inference of course depends on the actual hardware. When the toolbox is used in a cloud setting the utilization of the machine (if shared by multiple customers) might impact the measurements. Is this something that was empirically verified to be stable? Of course this also limits the type of devices that can be benchmarked on.
> >
> > To me the paper remains borderline in the end. But both other reviewers rate the paper highly. If the lack of insight from the actual study using this benchmark (actual proven value) should not be taken into consideration (which is unclear to me given this is a new track) then I have no issues with this paper being accepted.

---

> > > ### Comment · Reviewer_nd8M · 2021-07-21
> > > **Disagree with Reviewer #2**
> > >
> > > Respectfully, I do have to push back on some of this feedback. In your review, you tell the authors that they do not provide new insight with their case studies, but what *counts* as new insight? That commentary was vague. However, I agree with the authors that not showing novel insights into models made sense for the paper.
> > >
> > > Frankly, from a multiply marginalized perspective, I think it is dangerous business to assume the only or main contributions that should be made in the Datasets and Benchmarks track are theoretical contributions to machine learning methods because that is extremely inaccessible and one track minded. I don't understand the benefit to the overall community by making all authors provide novel insights into machine learning models, especially since authors may have different goals with their datasets/benchmarks. If I'm trying to demonstrate a novel dataset or benchmark, I don't think it's necessary to build the newest, craziest neural network to provide a 3% accuracy increase or something. I think running models in whatever capacity is appropriate for the problem domain is enough.
> > >
> > > You say that this benchmark does not have "actual proven value" when not only do they effectively argue that this benchmark fills a void with the current benchmark landscape (e.g., a benchmark that can satisfy NLP community needs), but they also go out of their way to provide a toolkit with a ton of features, many of which are novel. That is a contribution. Even more so, a benchmarking system cannot be understood in full until one understands how the community has benefited from it, and we cannot know that right now anyway since this paper just came out. That is, the people who really care about how this benchmark provides novel insight into models will be the users, and the authors had an appropriate scope for the paper in this respect.

---

### Official Review · Reviewer_nd8M · 2021-07-02
**Personalized Benchmarking with the Ludwig Benchmarking Toolkit**

**Rating:** 8
**Confidence:** 4

**Strengths:**

The authors don't explicitly mention this aspect although it is apparent, but their benchmark actually can highlight inequality and climate impact in a lot of ways because they provide a mechanism to show 'hey, their model performed better because they had a pod of TPUs at their trillion dollar tech company'. I think the note for a carbon footprint is extremely timely, especially considering Bender et al's paper at FAccT 2021 about the high carbon footprint of NLP models. I love that this paper also says 'let's go beyond accuracy!'. Obviously, there is more to a model than how well it optimizes for loss. In addition, there is a mechanism to keep track of criteria to potentially understand WHY one model outperforms another.

**Weaknesses:**

I would love to see an explanation of how one would account for differences in models where there are multiple variables at play. For example, X team had two different variables than Y team; which variable was most impactful, if any? Also, if the model is customized, how can one keep track of which of the criteria were impacted by such customization? How do we compare two different models that were not trained on the same datasets or don't use the same evaluation metrics? Or do we not?

As the authors point out, the benchmark only works on supervised tasks because of it's dependence on Ludwig, but to be fair evaluating unsupervised tasks is tricky and mostly non-standard. However, there is a growing interest in unsupervised deep generative and contrastive models, so it's still a weakness nonetheless.

**Additional Feedback:**

I think the paper solves a really good problem by providing control variables for some semblance of standardization across benchmarks.

**Clarity:**

I think the paper is well-written, but I would lowkey like to have seen some sections or statements appear earlier. However, this might be my reviewer hat rather than a general audience need.

One point of the paper where I'm a bit confused about is whether this is an NLP benchmark for text data or a generic benchmark for tasks across specialized AI domains, especially because RNN/LSTMs can be used for computer vision tasks. I noticed that the GitHub only included text dataset samples, but the paper does not explicitly refer to this benchmark as an NLP benchmark, whereas for example ImageNet is very explicitly a computer vision benchmark. It would be great if the authors could clarify about their vision of the scope of the benchmark in specialized AI tasks because the paper doesn't mention images in LBT until 5 pages in. The architectures that are provided are mostly used for NLP tasks in practice.

I'm also wondering what variety of tasks the fixed hyperparameters will perform well across different conditions (e.g., datasets, tasks, models)?

What scale is this benchmark measuring? There's a huge difference in the size of the models for a research paper using MNIST digits and an industry model that takes several months to train using an ungodly amount of hyperparameters. Another question is can this benchmark scale to ginormous models in industry or other places?

If a model performs great on this benchmark, is that the end all be all? What else should benchmarkers be looking for?

For some reason, Figure 1 in the paper shows MNIST and CIFAR-10 in a diagram of LBT, but I cannot find these as "default" datasets on the GitHub although I believe users have the option to add them on their own. However, it's kind of misleading, especially as the first graphic the reader sees.

Will there be no leaderboard whatsoever to show progress in the community, or will the results of the benchmark be internal, e.g. a company keeps track of their own internal leaderboard? I know the authors said this toolkit complements the leaderboards and that there is an Elasticsearch database. I also wonder if there's a way to log the submitters or if they can also stay anonymous?

I think there might be some missing or mislabeled figures that kind of threw off my ability to fully understand a few things??

I notice the authors bring up "robustness", but there isn't much discussion in the way of the relationship between accuracy, generalization, distributional shifts, etc. If you're going to talk about robustness, then a robust model goes beyond accuracy or even subgroup accuracy. So it seems LBT does not go beyond accuracy but rather is in addition to accuracy. I think there was a reference to adversarial examples, but the discussion on robustness was not unified at all. On that note, I don't know how the authors are defining robustness. While I sympathize with the page limit, it left me wanting. I think even the convenience that RobustnessGym already provides its own robustness metric, e.g. subpopulation performance, is a good reason to define robustness in a certain way. This was one of those things that made the paper not really self contained.

**Correctness:**

Are the claims made in the submission correct?
The authors are definitely biased toward their previous work and don't critique it the same way they do other works, but I didn't find any glaringly wrong claims.

If it is a benchmark, are the evaluation methods and experiment design appropriate and performed correctly?
Sure, there's more than one way to skin a cat.

**Documentation:**

Data Collection and Organization: The authors provide justification for how they chose their datasets based on diverse characteristics. They also discuss how they organize the data splits as control variables.

Ethical and Responsible Use: Neither the paper nor the GitHub outline an ethical and responsible use which is kind of disappointing, especially considering the work of racialized women on the topic of NLP+ethics within the past couple of years.

Availability and Maintenance: The benchmark toolkit is open source on GitHub (yay!). I assume the lab will maintain the benchmark and grow it, but I suppose that is up to the authors to decide. However, I don't think they have much of a choice since the environment variables are based on versioning of software, so it will have to be updated periodically. I believe part of the maintenance plan is the Elasticsearch database. I'm assuming the team will host it or give it to somebody else to host in the future.

I did not see a license on the GitHub. Is it creative commons?

I think there should be sufficient detail to support reproducibility with their experiments, yes.

**Ethics:**

Interestingly, the authors don't state any ethical considerations of their work. However, it is extremely important to understand that all data is social data. Humans can intentionally or unintentionally insert their own biases into datasets. For example, with language data, somebody could label offensive speech to a particular demographic as non-offensive either because of sheer ignorance or unconscious biases. Within the past year alone, benchmarks have been taken down and are no longer publicly available because they unknowingly scraped millions of images that were racist. Of course, this data is linguistic in nature, but the same principles apply. Data is not created out of thin air. It is inherently social. Finally, Bender et al. has a really good paper to reference for ethics of (large scale) NLP models, and there is more literature out there about it also, especially coming out of the NLP-focused conferences.

**Relation To Prior Work:**

The author(s) of the paper have also created DawnBench which is relatively well known. There is a website to compare models: https://dawn.cs.stanford.edu/benchmark/ImageNet/train.html. DawnBench uses the following criteria to compare models: Model	Time to 93% Accuracy, Cost (USD), Max Accuracy, Hardware, Framework (i.e., ML software). However, the only dataset is ImageNet, and thus measures accuracy for computer vision models only. Either way, they're pretty similar, but LBT has distinct contributions and takes it a step further. The authors cover other related work in the paper and provide justifications for how the work differs from previous contributions.

**Summary And Contributions:**

There's a huge problem with benchmarks in that there is no way to account for differences in performance among models viz. confounding variables. They respond to this issue with control variables, for example fixed hyperparameters, data splits, datasets, out of the box models, and so on. Even more so, the current benchmark community's focus on accuracy without considering other variables is insufficient to address benchmarking needs of the NLP and other communities. As such, the authors provide an array of evaluation metrics that go beyond performance, targeting what they term "value-drive communities".

---

> ### Author Response · Authors · 2021-07-12
> **Response to Reviewer #1 [Part 2/2]**
>
> 5. **Lack of support for unsupervised training**
>     We aim to support unsupervised training in a future iteration of LBT, as discussed in Section 5 "Discussion and Conclusion", and we appreciate your recognition of the growing relevance of such models
> 6. **Question about public leaderboard and sharing of experimental results**
>     At this time, we do not plan to host a public leaderboard. We see the main utility of LBT as enabling users to create personalized, multi-objective benchmarks representative of their objectives. This being said, we provide a hosted Elasticsearch database which can be used to view benchmark studies conducted by other users who have published their results to the database. To gain credentials to upload and explore experimental results in the shared instance, users must submit a Google Form via the main page of the GitHub repository. Users can choose to submit results to the database anonymously if they would like.
> 7. **Definition of Robustness**
>     In LBT, we define robustness as critical subpopulation based performance [1] and sensitivity to adversaries and input perturbations [2,3]. We acknowledge that there are other forms of robustness (like robustness to online distributional shift) that we are currently not addressing, and that can be the object of further extensions of the toolkit. In the revised manuscript, we clarify our definition of robustness by adding more details to the "Integrations for fine-grained evaluation" subsection in Section 3.3.
> 8. **Scale of models that can be benchmarked in LBT**
>     In our text classification study we demonstrate how LBT can be used to benchmark model performance across models trained from scratch, as well as large pretrained models which contain on the order of 60-120M parameters. In the future, we plan to support multi-gpu distributed training which will enable benchmarking for models with 1B+ parameters.
> 9. **Guidance to benchmarkers on what else they should be looking for**
>     One of the primary goals of LBT is to enable researchers and practitioners to design custom benchmarks. Thus, since we support a variety of benchmarking use cases, it is hard to provide a broad statement as to what benchmarkers should be looking for without having complete knowledge of their specific use cases. We leave it as future work to help practitioners understand what parameters to examine based on their specific task and believe that the shared experiment database will provide informative insights for such guidance.
> 10. **Maintenance and licensing of toolkit**
>     Thank you for requesting clarifications as to how the toolkit will be maintained. The toolkit will be maintained by the authors of the paper. Additionally, the Elasticsearch database will be hosted by the authors of the paper. All LBT source code is released under an Apache 2.0 license.
>
> Citations:
>
> [1] Goel, K., Rajani, N., Vig, J., Tan, S., Wu, J., Zheng, S., ... & Ré, C. (2021). Robustness gym: Unifying the nlp evaluation landscape. arXiv preprint arXiv:2101.04840.
>
> [2] Morris, J. X., Lifland, E., Yoo, J. Y., Grigsby, J., Jin, D., & Qi, Y. (2020). Textattack: A framework for adversarial attacks, data augmentation, and adversarial training in nlp. arXiv preprint arXiv:2005.05909.
>
> [3] Jia, R., & Liang, P. (2017). Adversarial examples for evaluating reading comprehension systems. arXiv preprint arXiv:1707.07328.

---

> > ### Comment · Reviewer_nd8M · 2021-07-21
> > **Good job!**
> >
> > I'm satisfied with the updates that were made to the paper based on all reviewer feedback. I think the paper is much better for the clarifications and additional details. Excited to see how people use this.

---

> ### Author Response · Authors · 2021-07-12
> **Response to Reviewer #1 [Part 1/2]**
>
> Thank you for your time and energy in reviewing our work. We are excited that you highlighted many of the key contributions of LBT such as enabling empirically sound benchmark comparisons by controlling for confounding variables and providing a framework for conducting multi-objective evaluations that allows different communities to benchmark models based on the criteria they care about. Moreover, your comment that another contribution of LBT is its ability to demonstrate the inequality and climate impact across different models was particularly encouraging as it identifies further use-cases of LBT beyond benchmarking. We appreciate your thoughtful comments and suggestions. Below, we respond to each comment in turn, while pointing out corresponding updates to the revised manuscript (additions are made in blue).
>
> 1. **Applicability of toolkit across AI domains**
>     We acknowledge the potential confusion regarding the scope of the benchmarking tasks and AI domains supported by LBT. We mention in the "Configurable" subsection in Section 3.3 that LBT can be used to construct benchmarking experiments across a large number of tasks including multi-class and multi-label classification, regression, sequence tagging, and sequence generation over a diverse set of input data types such as tabular, image, text, audio, and time series formats. We will make sure that this point is highlighted in the final manuscript.
>
>     To demonstrate LBT’s applicability to domains beyond NLP, we ran an additional benchmarking experiment comparing a Stacked CNN architecture (similar to VGG) against a ResNet-18 model on the MNIST and CIFAR10 datasets. The results and accompanying experiment configuration files can be found in the revised manuscript in "A.5 Additional Case Study: Image Classification Benchmark Experiment". Moreover, we have updated the GitHub repository to include both MNIST and CIFAR10 in the list of default datasets.
> 2. **Ethical considerations**
>     We deeply appreciate you raising concerns about the ethical considerations on the datasets and models used in LBT. We agree that such considerations are of critical importance and have thus introduced a section in our paper, Section 6 “Ethical Considerations”, that provides a more thorough discussion of the ethical considerations of our work.
> 3. **Performing feature and variable importance analyses**
>     We agree that having the ability to compute feature and variable importance analyses would be a great integration for the toolkit. A key challenge in performing these analyses is access to a standardized dataset which can be used to draw empirically accurate conclusions. LBT addresses this problem through its standardized training framework and shared Elasticsearch experiment database which helps users generate and share standardized data which can be used to perform variable importance analysis. At this time, LBT does not include any out-of-the-box integrations for feature importance and attribution analysis. However, in the future, we hope to integrate tools such as SHAP that compute feature importance analysis. Adding new evaluation/analyses tools to LBT is straightforward as demonstrated by the existing tooling integrations (e.g. TextAttack and Robustness Gym).
> 4. **Comparing models trained in different experiments**
>     Comparing models trained in different experiments is something that various users might be interested in. Fortunately, comparing the results of two experiments trained on different datasets can easily be done in LBT. Since LBT implements a standardized framework for collecting experiment metadata and metrics, it provides a common set of metrics which can be used to compare models from different experiments.

---

### Author Response · Authors · 2021-07-12
**Thank You and Summary of Key Paper Changes**

We thank the reviewers for their thoughtful comments. We were encouraged that they found the toolkit to be "directly aligned with the spirit of the NeurIPS benchmarks and datasets track" (R3) and the problem addressed by the toolkit to be relevant, critical, and underserved by existing practices. Moreover, we were excited that the reviewers strongly believe that the toolkit is useful for providing insights into "[why] one model outperforms another" (R1), highlighting "inequality and climate impact" (R1), benchmarking models "beyond accuracy!" (R1), standardizing benchmarks "by providing control variables for some semblance of standardization across benchmarks" (R1) and "[making] it easier for small research groups to carry out research in machine learning in a more principled and reproducible manner" (R3). We also value the feedback that the paper was well-written (R1, R2, R3) and that the design principles of the toolkit, including its configuration-based interface and "emphasis on measuring computational resource usage as well as accuracy", are valuable contributions and will "provide useful insight to others who wish to build on them." (R3). We appreciate the concerns of R2 that the case study lacked new insights, and would like to clarify that the main purpose of the case study was to demonstrate how various communities can derive value from the toolkit and how LBT satisfies these users’ unmet needs by enabling them to design configurable, standardized, and multi-objective benchmark studies. Please see our comment to R2 for a more detailed response.

We greatly appreciate all the questions and suggestions. This feedback has been helpful in improving the clarity and thoroughness of our work. All additions to the paper are made in blue. We address all reviewer specific questions and concerns in direct responses to the reviewers.

A summary of manuscript changes is as follows:

1. Additional Experiments
[Reviewer 1]
    - New "A.5 Additional Case Study: Image Classification Benchmark Experiment"
2. More Details on Metrics and Models
[Reviewer 2]
    - New subsection "Metric Details" in A.1 Metrics
    - New "A.3 Off-the-shelf Models"
3. Ethical Considerations
[Reviewer 1]
    - New Section 6 "Ethical Considerations”
4. Lessons Learned
[Reviewer 3]
    - New "A.6 Lesson Learned"
5. Clarification of Definition of Robustness
[Reviewer 1]
    - Additions to "Integrations for fine-grained evaluation" subsection in Section 3.3
6. Clarification of Intent of Text Classification Case Study
[Reviewer 2]
    - Additions to Section 4 "Case Study: Large-Scale Text Classification Analysis"

---

### Decision · Program_Chairs · 2021-07-26

**Decision:**

Accept

**Comment:**

The paper provides a software framework for conducting machine learning benchmarks. The reviewers agree that such software will be useful for the community, and that the paper is of high quality. The main concern is whether a paper on software, as opposed to a dataset or benchmark, is in scope for the datasets & benchmarks track. Due to the importance of open source software for machine learning research, I lean towards a broad interpretation of the datasets & benchmarks track to include software for benchmarking, and hence recommend to accept the paper.